# Bioinformatic Characterization and Molecular Evolution of the *Lucina pectinata* Hemoglobins

**DOI:** 10.3390/genes13112041

**Published:** 2022-11-05

**Authors:** Ingrid M. Montes-Rodríguez, Carmen L. Cadilla, Juan López-Garriga, Ricardo González-Méndez

**Affiliations:** 1Cancer Biology Division, PROMIC, Comprehensive Cancer Center of the University of Puerto Rico, San Juan, PR 00936-3027, USA; 2Department of Biochemistry, School of Medicine, University of Puerto Rico-Medical Sciences Campus, San Juan, PR 00936-5067, USA; 3Department of Chemistry, Faculty of Arts and Sciences, University of Puerto Rico—Mayagüez Campus, Mayagüez, PR 00681-9000, USA; 4Department of Radiological Sciences, School of Medicine, University of Puerto Rico-Medical Sciences Campus, San Juan, PR 00936-5067, USA

**Keywords:** *Lucina pectinata*, *L. pectinate*, hemoglobins, sulphur-binding hemoglobins, bioinformatics, phylogenetics of hemoglobins in mollusks

## Abstract

(1) Introduction: *Lucina pectinata* is a clam found in sulfide-rich mud environments that has three hemoglobins believed to be responsible for the transport of hydrogen sulfide (HbI_Lp_) and oxygen (HbII_Lp_ and HbIII_Lp_) to chemoautotrophic endosymbionts. The physiological roles and evolution of these globins in sulfide-rich environments are not well understood. (2) Methods: We performed bioinformatic and phylogenetic analyses with 32 homologous mollusk globin sequences. Phylogenetics suggests a first gene duplication resulting in sulfide binding and oxygen binding genes. A more recent gene duplication gave rise to the two oxygen-binding hemoglobins. Multidimensional scaling analysis of the sequence space shows evolutionary drift of HbII_Lp_ and HbIII_Lp_, while HbI_Lp_ was closer to the *Calyptogena* hemoglobins. Further corroboration is seen by conservation in the coding region of hemoglobins from *L. pectinata* compared to those from *Calyptogena*. (3) Conclusions: Presence of glutamine in position E7 in organisms living in sulfide-rich environments can be considered an adaptation to prevent loss of protein function. In HbI_Lp_ a substitution of phenylalanine in position B10 is accountable for its unique reactivity towards H_2_S. It appears that HbI_Lp_ has been changing over time, apparently not subject to functional constraints of binding oxygen, and acquired a unique function for a specialized environment.

## 1. Introduction

Invertebrate hemoglobins (Hbs) display higher variability in terms of amino acid chain constitution, quaternary structures, and functional properties when compared to their vertebrate counterpart [1]. Some are monomeric, while others can form large polymers. In invertebrates, Hbs can be expressed differentially in different anatomical sites, and show differences in their ligand kinetics [1]. Since invertebrates can be found in a wide range of habitats, the variation encountered in their Hbs is indicative of how these molecules must adapt to fulfill their function regardless of the environmental challenges in the surrounding areas where they live [2].

The globin fold is a well characterized protein structure, first described for sperm whale myoglobin (Mb) and horse oxyhemoglobin, composed of up to eight α helices, labeled A to H, and residues within each helix segment are numbered from the amino end [3,4,5]. Perutz [6] established structural similarities between various globins and sperm whale myoglobin that allows the use of the same notation for equivalent structural residues in all hemoglobins, regardless of additions or deletions in their primary structure. Bashford, Chothia, & Lesk [7] postulated that at least 32 internal conserved regions, possessing mostly hydrophobic residues, define the overall globin fold structure. Lesk & Chothia [8] demonstrated that the differences in amino acid sequences between these molecules lead to the formation of a well conserved globin structure by a series of low-energy mutations producing limited structural changes and by the functional advantage of any modulation of the ligand affinity caused by these changes.

In sulfide-rich habitats, Hbs are essential for the sustainability of the chemoautotrophic based symbiosis found in many invertebrates [9,10]. The hosts supply the symbionts with inorganic substrates, such as sulfide, while the symbionts are believed to provide the host with autotrophically fixed carbon [11]. A remarkable aspect of the Hbs of these organisms is their ability to retain their function in presence of hydrogen sulfide (H_2_S) without the covalent modification of the heme pyrrole rings that leads to formation of sulfhemoglobin, as has been seen in most mammalian myoglobins (Mbs) and Hbs [12]. However, the biochemical characteristics of the Hbs from these species differ substantially.

Chemosymbiosis with sulfide-oxidizing bacteria is found in members of the *Lucinidae* family, which have an extensive variety of species and are found across the globe [13]. The clam *Lucina pectinata* is found in the southwest coast of Puerto Rico and throughout the Caribbean Sea. As it lives in sulphide-rich muds, it has developed a symbiotic relation with intracellular bacteria for which the clam provides both hydrogen sulfide (H_2_S) and oxygen (O_2_). Three different Hbs have been identified in *L. pectinata*: hemoglobin I (HbI_Lp_), a sulfide-reactive hemoglobin, and hemoglobin II (HbII_Lp_) and hemoglobin III (HbIII_Lp_), which are oxygen-reactive hemoglobins [14].

The sulfide binding HbI_Lp_ is a monomeric globular protein, composed of 143 amino acid residues [15]. HbI_Lp_ has a conventional globin fold lacking the D helix. This is similar to plant and truncated hemoglobins [16]. “HbI_Lp_ is characterized by an unusual distribution of phenylalanines at the heme distal positions B10, CD1 and E11” [17]. HbII_Lp_ and HbIII_Lp_ (150 and 153 amino acid respectively) have a tyrosine residue at position B10 [18] (Figure 1 below). Structural alignment of HbI_Lp_ protein structure (PDB 1FLP) and Sperm Whale Myoglobin (SWM) (PDB 1A6M) showing their amino acid distribution in the heme pocket. For SWM there is a leucine at position B10, a histidine at position E7 and a valine in position E11. (see Appendix A).

Amino acid sequences of HbII_Lp_ and HbIII_Lp_ have a 65% of identity and they both have lower sequence identity with HbI_Lp_ (32% of identity with HbII_Lp_ and 33% of identity with HbIII_Lp_). In these three hemoglobins the heme distal E7 position has a glutamine residue instead of the typical distal histidine (Figure 1 shows the comparison of the heme pockets of HbI_Lp_ and the sperm whale myoglobin). These three Hbs have been extensively studied in order to understand their chemistry, the factors that affect their specificity, and they have been used as models to understand distal ligand binding control [19,20,21,22,23]. How the environment influences the evolution of function in these hemoglobins has yet to be elucidated. We compared the *L. pectinata* Hbs to other homologous globin sequences from mollusks to identify conserved features and evolutionary relationships. We also compared the coding sequences of *L. pectinata* Hbs between them and with those available for *Calyptogena* spp. in order to obtain a better understanding of globin gene structure and sequence conservation in mollusks.

## 2. Materials and Methods

### 2.1. Bioinformatic Analyses

The sequences of *L. pectinata*’s Hbs (mRNA: HbI_Lp_-AF187049; HbII_Lp_-AY243364; HbIII_Lp_-EU040120; protein: HbI_Lp_ -AAG01380; HbII_Lp_-AAO89499; HbIII_Lp_-ABS87592) were retrieved from the National Center for Biotechnology Information (NCBI) GenBank and GenPept databases [24]. Retrieval of sequences homologous to the three protein sequences of the *L. pectinata* hemoglobins was performed with the Basic Local Alignment Search Tool (BLAST) [25,26,27] using blastp with default parameters and selecting to search only in the mollusks (taxonomic id 6447). Thirty-two unique protein sequences were retrieved from the three BLAST homology searches with an expectation value threshold smaller than 0.0001 for significance and/or sequence similarity of approximately 25% or greater. The accession numbers of these sequences together with the BLAST search result summaries are shown in Appendix A.

Multiple sequence alignment (MSA) was performed with globin sequences homologous to HbI_Lp_, HbII_Lp_ and HbIII_Lp_. The MSA includes globins from the deep-sea clam species of *Calyptogena*, marine gastropods from *Aplysia* spp., ark clams, also known as blood clams (*Anadara* spp., *Barbatia* spp., *Scapharca* spp. and *Tegillarca* spp.), the Atlantic surf clam *Spisula solidissima* and the Antarctic bivalve *Yoldia eightsi*. The *Physeter catodon* (Sperm Whale) myoglobin amino acid sequence (Accession number NP_001277651) and the helical segments of its globin three-dimensional structure (Accession number 1EBC_A) were included in the MSA as reference, to aid identification of amino acid residues based on the myoglobin fold [7,8] and the standard globin fold nomenclature [6].

Multiple sequence alignment (MSA) of these 32 homologous sequences was performed using the PSI-COFFEE alignment program. This algorithm is used to align distantly related proteins using the consistency function and profile information with homology extension [28]. Given the low sequence similarity, around the so-called ‘twilight zone’ for detection of homology, this is the method of choice for aligning these sequences. These results were visualized using GeneDoc [29].

The structural alignment of the protein structures of HbI_Lp_ (PDB:1FLP) and HbII_Lp_ (PDB:2OLP), and HbI_Lp_ (PDB:1FLP) and Sperm Whale Myoglobin (SWM) (PDB:1A6M), showing the spatial orientation and distribution of amino acids in the heme pocket was performed using the MultiSeq bioinformatics analysis environment [30] using the Structural Alignment of Multiple Proteins algorithm [31] and the RMSD function in the VMD software [32]. These comparisons help illustrate the overall structural conservation of the hemoglobin fold and the heme O_2_/H_2_S binding pockets.

### 2.2. Phylogenetic Analysis

Phylogenetic analysis of these globin sequences was performed as follows. The MSA of the 32 amino acid sequences was trimmed using trimAl with the automated heuristic setting *‘automated 1’* [33]. The trimmed MSA was then converted to the PHYLYP format. Using the PHYLIP suite of programs [34], an unrooted Maximum Likelihood phylogenetic tree with 1000 bootstrap replicates (SEQBOOT) was constructed using the protein maximum likelihood routine (ProML) in its parallel implementation (MPI-ProML) [35] using 16 processors in the BioU computer cluster at the Pittsburgh Supercomputing Center. A consensus phylogenetic tree was generated with the CONSENSE routine using the extended majority rule and visualized with FigTree [36]. A rooted species phylogenetic tree was built using the Taxonomy resource at NCBI [37,38] and was verified using the Tree-of-Life resource [39]. The gene tree and the species tree were then reconciled using the program NOTUNG [40,41] and a root was placed on the globin gene tree using the rooting procedure in NOTUNG [40,41,42,43]. Visualization of the reconciled phylogenetic trees was done with NOTUNG routines that are based on the ATV routine in FORESTER [44].

### 2.3. Metric Multidimensional Scaling Analysis

The trimmed MSA was used as input for metric multidimensional scaling (MMDS) analysis and exploration of the sequence space. The MMDS analysis was performed with the R package *bios2mds* [45], using a JTT similarity matrix to construct the distance matrix required for the analysis. A clustering analysis was performed as part of the MMDS using the routines in the package and optimized based on a Silhouette analysis. These were performed as described in [45,46].

### 2.4. Coding Region Alignments

Multiple Sequence Alignments (MSA) of the coding sequences of the three *L. pectinata* hemoglobin sequences were constructed using M-COFFEE [47]. Motif analysis was done using the Maximum Entropy Motif Elicitation program (MEME) in the zero-or-one-per-sequence mode, requesting 6 motifs [48]. All the alignment results were visualized using the program GeneDoc [29].

Retrieval of sequences homologous to the mRNA sequences of the *L. pectinata* Hbs was performed with BLAST [25,26,27] using blastn with default parameters and optimized for somewhat dissimilar sequences. The only sequences that showed homology with a significant expectation value were the *L. pectinata* hemoglobin sequences for HbII_Lp_ and HbIII_Lp_, and that homology was with each other. HbI_Lp_ did not retrieve any nucleotide sequence. The sequences of coding regions of *Calyptogena spp*. Hbs (*C. nautilei* HbIV-AB186050.1; *C. nautilei* HbIII-AB186049.1; *C. soyoae* HbI-AB186047.1; *C. soyoae* HbII-AB186048.1; *C. tsubasa* HbI-AB186401.1; *C. tsubasa* HbII-AB186402.1; *C. kaikoi* HbI-AB186045.1; *C. kaikoi* HbII-AB186046.1) were also retrieved from the NCBI GenBank. MSA, MEME analysis and visualization was performed as indicated in the previous paragraph.

## 3. Results

### 3.1. Multiple Sequence Alignment

The MSA shows a high quality alignment confirmed by the conservation of the invariant proximal histidine at position F8 (His-F8), the highly conserved phenylalanine at position CD1 (Phe-CD1), and the phenylalanine at position B14 (Phe-B14) (Figure 2). A tryptophan at position H8 replaces the methionine found in sperm whale myoglobin. This is an amino acid replacement found in invertebrate Hbs [7,49]. The conservation of key positions in hemoglobins can be explained by the previous work of Ptitsyn and Ting [50] with the analysis of sequences from twelve globin subfamilies. In this analysis, Ptitsyn and co-workers showed that there are two common conserved clusters of amino acids between these globins. The first cluster includes residues in direct contact with the heme (CD1, F8, E11, FG5, F4, and G5), and heme neighboring residues (C2, B14, B10, E4, CD4, H19 and B13). The second cluster incudes hydrophobic residues distant from the heme center that may play an important role in the first steps of the protein folding process (A8, A12, G12, G16, H8 and H12) [50]. Additionally, the work of Liong et al. [51] with sperm whale myoglobin showed the relevance of residues at positions F4, FG3, FG5, and G5, in retaining the porphyrin prosthetic group. They demonstrated that the residues in positions Leu-F4, His-FG3, Ile-FG5, and Leu-G5 create a hydrophobic environment around the porphyrin preventing hydration of the heme. In our analysis, positions A12, B6, B13, B14, E4, FG5 and H8 show a high level of conservation (100% or >80% conservation level) presenting either the same residue or residues with the same physicochemical properties. The positions A8, B10, C2, CD4, E11, F4, G5, G12, G16, H12 and H19 have lower conservation (>50% conservation level), or no conservation among the sequences.

### 3.2. Phylogenetic Analyses

Figure 3 shows the results of our phylogenetic analyses. A maximum likelihood (ML) gene tree constructed with 32 mollusk globin sequences is shown in Figure 3A. The un-rooted ML gene tree shows three major clades: One clade corresponds to the Heterodonta subclass grouping the globins of *L. pectinata* with the *Calyptogena* spp. and *S. solidissima*. Another clade corresponds to those clams belonging to the Pteriomophia subclass grouping the globins of *Barbatia* spp., *Tegillarca* spp., *Anadara* spp., and *Scapharca* spp. A third clade clusters the members of the subclass Opisthobranchia, the myoglobins of the gastropods *Aplysia* spp.; *Y. eightsi*, a clam that belongs to the subclass Protobranchia is shown in black and does not cluster with any group. The rooted-species-reconciled gene tree is shown in Figure 3B. The species tree (Figure 3C), even though it is polytomic, has a well-defined root between the Gastropoda (*Aplysia* spp.) and the Bivalvia (all other groups in our analysis).

The ML phylogenetic tree divides these globin sequences in three main clusters: Bivalves belonging to the Heterodonta subclass; Bivalves belonging to the Pteriomophia subclass; and the Gastropods belonging to the Opisthobranchia subclass. We can group the globin sequences according to the clade that they belong to, and examine the amino acid conservation within each group (Figure 4).

We will discuss the similarities of those organisms that live in environments similar to that of *L. pectinata*, where the levels of hydrogen sulfide are high. The members of the *Calyptogena spp*., as well as *L. pectinata*, rely on the symbiosis with chemoautotrophic sulfur-oxidizing bacteria for their nutrition [52,53,54,55,56] and live in reduced environments where low to moderate concentrations of hydrogen sulfide are found, i.e., cold seeps and hydrothermal vents [57]. There are a few positions with conserved residues in this group that are different from the other groups: the distal heme positions Gln-E7, Phe-E11, position Phe-B9, and the B10 position, occupied mainly by a tyrosine, with the exception of HbI_Lp_ from *L. pectinata* that has phenylalanine. The topological positions B10, E7 and E11 are shown in the structural alignments of HbI_Lp_ vs. HbII_Lp_ (Figure 1) and HbI_Lp_ vs. SWM (Appendix A).

Table 1 shows amino acids in positions B9, B10, E7 and E11 for this subgroup. The nerve Hb of *S. solidissima*, which is grouped in the Heterodonta clade with the Hbs from *L. pectinata* and *Calyptogena* spp., has His-E7 and inhabits sandy continental shelf habitats. Hence, this organism was not included when we looked for similarities between organisms living in sulfide-rich environments.

### 3.3. Metric Multi-Dimensional Scaling Analysis

The Metric Multi-Dimensional Scaling analysis (MMDS) was performed with the trimmed alignment of the 32 mollusk globin sequences creating a multi-dimensional sequence space, with a 3D projection of major principal components shown in Figure 5.

This type of analysis is a powerful method used to visualize distance between sequences and complements phylogeny providing evolutionary information [46]. This sequence space is defined by the principal components (PCs). In Figure 5 we show the 3-D projection of the first three PCs. It is highly informative about the evolution and drift of these proteins. In this analysis we observed, as expected, that these globins are also clustered according to their subfamilies. However, the 3D sequence space clearly shows how HbII_Lp_ and HbIII_Lp_ are close to each other and separated from the other members of the Heterodonta subfamily. On the other hand, HbI_Lp_ is closer to other members of the Heterodonta subfamily despite its unique hydrogen sulfide reactivity. This data shows that HbII_Lp_ and HbIII_Lp_ have drifted evolutionarily in the sequence space. The 2D projections for pairs of these three PC are also shown in Appendix A as B, C and D. In this analysis we observed, as expected, that these globins are also clustered according to their subfamilies. The 2D sequence space of PC1 and PC2, Appendix A, resembles the unrooted phylogenetic tree previously shown. However, the 3D sequence space clearly shows how HbII_Lp_ and HbIII_Lp_ are close to each other and separated from the other members of the Heterodonta subfamily. This can also be seen in the other two 2D sequence space shown in Appendix A. These results revealed that even though HbII_Lp_ and HbIII_Lp_ are O_2_ transporting Hbs like the ones found in the *Calyptogena* spp., they are evolving independently as an outgroup while the HbI_Lp_ sequence conserves a higher degree of evolutionary similarity to the *Calyptogena* spp. Hemoglobins.

### 3.4. Coding Region Bioinformatic Analyses

The level of conservation between the three coding region sequences of HbI_Lp_, HbII_Lp_ and HbIII_Lp_ of *L. pectinata* is presented in the multiple sequence alignment in Figure 6A.

The three Hbs from *L. pectinata* have intron insertions at the conserved positions B12.2 and G7.0 [58], indicated by the arrows in Figure 6A that mark these intron-insertion positions, respectively (B12.2 is the intron inserted after the second base of the codon of amino acid 12 on the B-helix and G7.0 is the intron inserted between the codons of amino acids 6 and 7 on the G-helix). For the three coding regions we observed high conservation surrounding the regions of intron insertion at the B12.2 position (after nucleotides 95, 98 and 101 in the coding region of HbI_Lp_, HbII_Lp_ and HbIII_Lp_, respectively), while there is more variability in the region of the third intron insertion at position G7.0 (after nucleotides 324, 327 and 330 in the coding regions of HbI_Lp_, HbII_Lp_ and HbIII_Lp_, respectively). Overall, the region before the second and third intron insertion is more conserved between the three Hb, while the region after the intron insertion at G7.0 is more conserved between HbII_Lp_ and HbIII_Lp_, but not for HbI_Lp_. This variability is corroborated at the protein level where motif analysis using MEME showed three major motifs (see Figure 6B), two of them are in the three Hbs (blue and red motifs), while the last motif (colored in gray) is not detected in HbI_Lp_.

Motif analysis of the hemoglobin coding regions from L. pectinata and the genus Calyptogena showed that two out of six motifs, colored in brown and light-gray, are shared between all of the members of the Heterodonta clade (see Figure 7).

However, for the HbI_Lp_ coding region we find a weak signal for the motifs shared with the *Calyptogena spp.* sequences, indicated in yellow, pink and blue, with *p*-values of 7.99 × 10^−5^, 6.39 × 10^−5^, and 3.01 × 10^−5^ respectively. The HbII_Lp_ coding region also shows a weak signal for a motif with a *p*-value of 1.23 × 10^−5^, indicated in blue.

## 4. Discussion

Bioinformatic analyses between *L. pectinata* hemoglobins and globin sequences from other mollusks living in sulfide-rich environments revealed sequence conservation in the B10, E7, and E11 residues that give these proteins a unique distal site structure allowing them to carry their normal function in such extreme environments. The mollusk globin sequences used for this analysis differ in terms of subunit structure and ligand specificities. For example, HbI_Ck_ and HbII_Ck_ from *C. kaikoi* can assume homodimeric or homotetrameric conformation in a concentration dependent manner. Since the amino acid sequence of HbI_Ck_ and HbII_Ck_ are identical to MbI_Ck_ and MbII_Ck_, respectively, found in this clam’s muscle tissue, it has been suggested that they may serve as oxygen storage molecules [59]. *C. nautilei* hemoglobins HbIII_Cn_ and HbIV_Cn_ are monomeric hemoglobins while hemoglobins HbI_Ct_ and HbII_Ct_ from *C. tsubasa* form dimers [60]. In the case of *C. soyoae*, it has two homodimeric hemoglobins, HbI_Cs_ and HbII_Cs_, and for HbII_Cs_ the autoxidation rate under physiological temperature is three times slower than the one observed for human hemoglobin, which, in combination with its relatively high O_2_ affinity, is indicative that it can transport O_2_ under physiological conditions [53,54]. On the other hand, globins from blood clams tend to form dimers and tetramers that bind oxygen cooperatively [57,61,62]. Bao et al. [63] described a new role for the Hbs from the blood clam *T. granosa*, since they are involved in the immune defense after the clams were exposed to bacterial infection. Dewilde and collaborators [64] concluded that the nerve hemoglobin (nHb) from the surf clam *S. solidissima* has a myoglobin-like function since it has a moderate O_2_ affinity and phylogenetic analysis with mollusks Hb and vertebrate nHb placed this globin with mollusks [65]. Finally, the Antartic clam *Y. eightsi* possesses a hemoglobin with O_2_ affinity similar to that of mammalian Hb, at the clam’s physiological temperature. This affinity decreases significantly at 25 °C, suggestive of this clam’s adaptation to a low temperature environment [66].

We looked for similarities between the globins from members of the *Calyptogena* spp. and *L. pectinata* since both organisms live in environments that have high concentration of H_2_S. For the set of mollusk globin sequences in this study, they show clade-specific conservation or structurally conservative substitutions. The positions with residues conserved in this group are the Gln-E7, Phe-E11, and the Tyr-B10 (with the exception of HbI from *L. pectinata* that has Phe-B10) and position Phe-B9. The globins from members of the Heterodonta subclass and *Y. eightsi* have a phenylalanine at E11, while the rest of the globin sequences possess an aliphatic residue (Val, Ile, or Leu). Several studies have demonstrated the importance of the residue occupying the E11 position since it is part of an overall kinetic barrier for the ligand in the heme pocket [67,68]. The residue at position E7 is considered to be essential for the stabilization of the bound ligand through hydrogen bonding, serves as a gate for ligand entry, and plays a crucial role in the prevention of the autoxidation of the heme [66,69,70,71]. The stabilization of ligands by hydrogen bonding with glutamine in the E7 position has been described for the three *L. pectinata* Hbs [17,21,72] Previous work where the HbI_Lp_ heme pocket was mutated to mimic the human myoglobin distal site demonstrated that His-E7, in the presence of H_2_S and H_2_O_2_ or O_2_, is essential for the formation of sulfhemoglobin, where sulfide is covalently bound in the pyrrole of the porphyrin ring resulting in a non-functional hemoglobin [73]. Interestingly, polymeric-hemoglobin from symbiont-containing tubeworms living in deep-sea hydrothermal vents contain histidine at the E7 position [56] and are capable of binding simultaneously and reversibly O_2_ (in their heme pockets) and H_2_S by the formation of zinc-sulfide complexes with Zn ions that are bound to specific residues (B7, B12, B16, CD10, G9, G13 and G16) across the globin chain [74,75]. This zinc-binding function has not been described in the *Lucinidae* spp. or *Calyptogena* spp. hemoglobins.

We previously mentioned some studies that described the role of several positions across the globin structure and observed that some positions showed less conservation across the mollusk globin sequences in our analysis. Some of these positions correspond to residues located at the immediate distal heme pocket vicinity. Studies have shown that the variability in ligand affinity and autoxidation of these globins are mostly determined by the physicochemical properties of the residues occupying positions at the distal heme pocket. In their review, Springer et al. [66] discussed myoglobin ligand binding recognition in studies where site-directed mutagenesis of distal residues was combined with their structural and physicochemical characterization. Based on that review, it becomes clear that a combination of steric hindrance and polarity of the distal pocket governs hemoglobin ligand recognition, affinity and autoxidation. The positions B10, E11 and E7 of the distal heme environment play a predominant role in the control of ligand binding kinetics [76]. One of the most relevant similarities between these globins is the invariant presence of a glutamine residue in the E7 position. As mentioned before, the existence of a histidine residue in this particular position will promote the formation of sulfhemoglobin in the presence of H_2_S and O_2_ or H_2_O_2_, resulting in a nonfunctional protein. This single amino acid substitution may be considered an evolutionary adaptation of these organisms in order to increase their fitness in these sulphide-rich environments.

Like HbII_Lp_ and HbIII_Lp_ from *L. pectinata*, the hemoglobins from *Calyptogena* spp. included in this study bind and transport O_2_ in the presence of H_2_S and their identical distal site residues distribution may imply a similar binding mechanism. The main difference in the distal cavity between the sulfide reactive hemoglobin from *L. pectinata*, HbI_Lp_, and the oxygen reactive hemoglobins is a single substitution of a phenylalanine in the B10 position instead of a tyrosine, which has been demonstrated to be accountable of HbI_Lp_’s unique reactivity towards H_2_S [77]. HbII_Lp_ and HbIII_Lp_ have high O_2_ affinities and they remain oxygenated in the presence of H_2_S [78,79] and their O_2_ dissociation rates are three times slower than dissociation from HbI_Lp_ and very slow compared when compared with other Hbs [14]. In presence of H_2_S, HbI_Lp_ forms a ferric hemoglobin sulfide [14]. Studies with other hemoglobins that also have Gln-E7 and Tyr-B10 in their heme pockets, revealed that Tyr-B10 directly interacts with the bound O_2_, forming a hydrogen bond network between Gln-E7, B10 Tyr, and O_2_, that accounts for the low O_2_ dissociation rate of these hemoglobins [80,81,82]. Several studies have shown how Phe-B9 affects the orientation of the aromatic residues in the positions B10 and E11 due to electrostatic aromatic interactions [17,83]. Therefore, comparing the hemoglobin sequences from *L. pectinata* to sperm whale myoglobin allows presenting that a single amino acid in the heme proteins: Gln-E7 leads to transport of hydrogen sulphide in *L. pectinata* hemoglobin I, while His-E7 leads to the modification of the heme group to generate sulfmyoglobin. Thus, a single amino acid substitution can lead to differences in these heme proteins’ function. Studies with the sperm whale myoglobin triple mutant (LeuB10Phe, HisE7Gln, and ValE11Phe), mimicking the HbI_Lp_ distal environment, showed a change in orientation, not only in Phe-B10 but also for Phe-E11, which assume a more perpendicular position to the heme plane, since the B9 position is a isoleucine residue, resulting in destabilization of the bound ligand [84].

The phylogenetic analysis shows a consistent and robust gene tree. Based on these analyses, we suggest that a first gene duplication at an early evolutionary stage resulted in the ancestor of the HbI_Lp_ gene and an ancestor of the HbII_Lp_/HbIII_Lp_ gene. A more recent gene duplication of the oxygen-binding gene occurred to give rise to genes for two oxygen-binding proteins, HbII_Lp_, and HbIII_Lp_. This hypothesis is supported with the fact that HbI_Lp_ not only showed lower sequence similarity to both HbII_Lp_ and HbIII_Lp_ but also has different ligand kinetics. Furthermore, this corroborates the similarities at the sequence level and kinetic mechanisms between HbIILp and HbIII_Lp_. In addition, HbI_Lp_ appears to have been changing over time, apparently not subject to the constraint of binding oxygen, acquiring a unique function for a specialized environment.

Metric Multi-Dimensional Scaling analysis of mollusk hemoglobins showed evolutionary drift of HbII_Lp_ and HbIII_Lp_, while HbI_Lp_ was closer to the *Calyptogena* spp. hemoglobins in sequence space. The MMDS shows that HbII_Lp_ and HbIII_Lp_ have drifted evolutionarily in the sequence space. These results revealed that even though HbII_Lp_ and HbIII_Lp_ are O_2_ transporting Hbs like the ones found in the *Calyptogena* spp., they are evolving independently as a separate subgroup while the HbI_Lp_ sequence maintains a higher degree of similarity to the *Calyptogena* spp. hemoglobins.

The coding region analysis of hemoglobins from *L. pectinata* with those from *Calyptogena* spp., suggest that HbI_Lp_ from *L. pectinata* has features that are shared with the *Calyptogena* spp. hemoglobins. These similarities are supported by previous work by Suzuki and collaborators, who proposed that the *L. pectinata* and *Calyptogena* spp. hemoglobin chains evolved from a common ancestor [59]. The conservation of intron locations interrupting the coding region of their cognate genes is further evidence of the conservation of globin gene structure in invertebrate hemoglobins.

## 5. Conclusions

Bioinformatic analyses between *L. pectinata* hemoglobins and globin sequences from other mollusks living in sulfide-rich environments revealed what appears to be clade-specific sequence conservation in residues that give these proteins a unique distal heme pocket site structure that allows for normal function in such extreme environments. One of the most relevant similarities between these globins is the invariant presence of a glutamine residue in the E7 position. This single amino acid substitution may be considered an evolutionary adaptation of these organisms in order to increase their fitness in these sulphide-rich environments. Phylogenetics suggests that in *L. pectinata* a first gene duplication resulting in sulfide binding and oxygen binding genes. A more recent gene duplication gave rise to the two oxygen-binding hemoglobins. Metric Multi-Dimensional Scaling analysis of mollusk hemoglobins showed evolutionary drift of HbII_Lp_ and HbIII_Lp_, while HbI_Lp_ was closer to the *Calyptogena* spp. hemoglobins in sequence space. This is further corroborated by the conservation seen in the coding region of hemoglobins from *L. pectinata* with those from *Calyptogena* spp.

## Figures and Tables

**Figure 1 genes-13-02041-f001:**
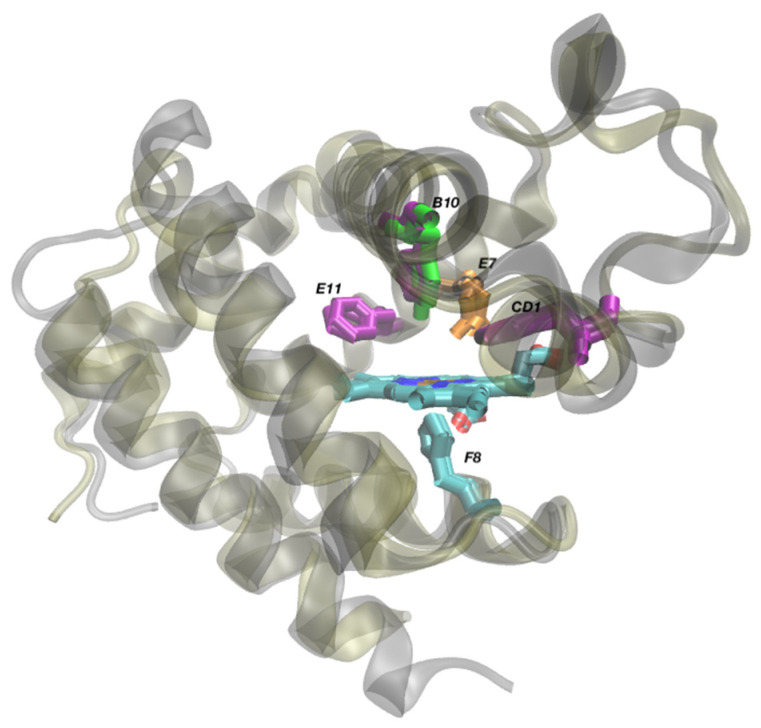
Structural alignment of HbI_Lp_ protein structure (PDB 1FLP) and HbII_Lp_ (PDB 2OLP) showing their amino acid distribution in the heme pocket. Alignment was done using the STAMP algorithm as implemented in MultiSeq in VMD. The only difference between the two heme pockets is the amino acid in position B10, a phenylalanine for HbI_Lp_ and a tyrosine for HbII_Lp_. Both hemoglobins have a phenylalanine at positions CD1 and E11, a glutamine at position E7 and a histidine at position F8. HbII_Lp_ and HbIII_Lp_ have the same heme pocket amino acid distribution. The RMSD for the backbones (residues 2–147) is 6.54 angstrom.

**Figure 2 genes-13-02041-f002:**
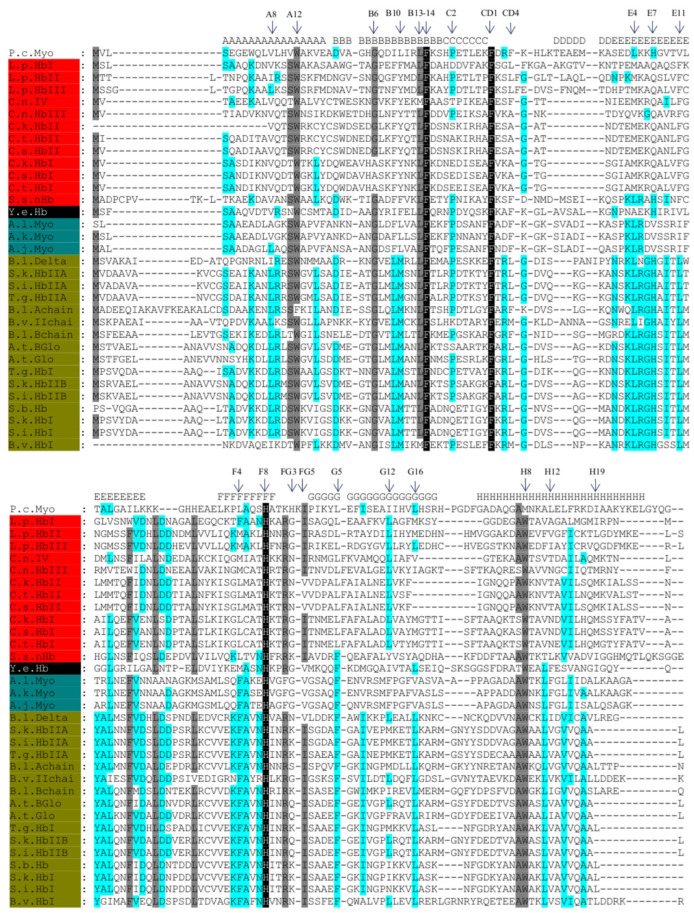
GeneDoc visualization of Multiple Sequence Alignment of 32 mollusk protein sequences using PSI-COFFEE. The conservation is shown in three levels based on similarity groups: Black for 100% conservation, Gray for 75% of conservation and light blue for 50% conservation. The abbreviations are the following: L.p (*L. pectinata*), C.n. (*C. nautilei*), C.k. (*C. kaikoi*), C.t. (*C. tsubasa*), C.s. (*C. soyoae*), B.l. (*Barbatia lima*), S.s. (*Spisula solidissima*), Y.e. (*Y. eightsi*), A.l. (*Aplysia limacine*), A.k. (*Aplysia kurodai*), A.j. (*Aplysia juliana*), S.k. (*Scapharca kagoshimensis*), S.i. (*Scapharca inaequivalvis*), T.g. (*Tegillarca granosa*), B.v. (*Barbatia virescens*), A.t. (*Anadara trapezia*), S.b. (*Scapharca broughtonii*), and P.c. (*P. catodon*). On top of the alignment the secondary structure of *P. catodon* myoglobin is indicated, according to its crystallographic structure (accession number 1EBC_A). The organism identifiers are shaded based on the clade division of phylogenetic analysis: red indicates the Bivalves belonging to the Heterodonta subclass; in ‘olive’, we indicate the Bivalves belonging to the Pteriomophia subclass; blue was used for the Gastropods belonging to the Opisthobranchia subclass; and in black the Bivalve, *Y. eightsi*, a member of the Protobranchia subclass.

**Figure 3 genes-13-02041-f003:**
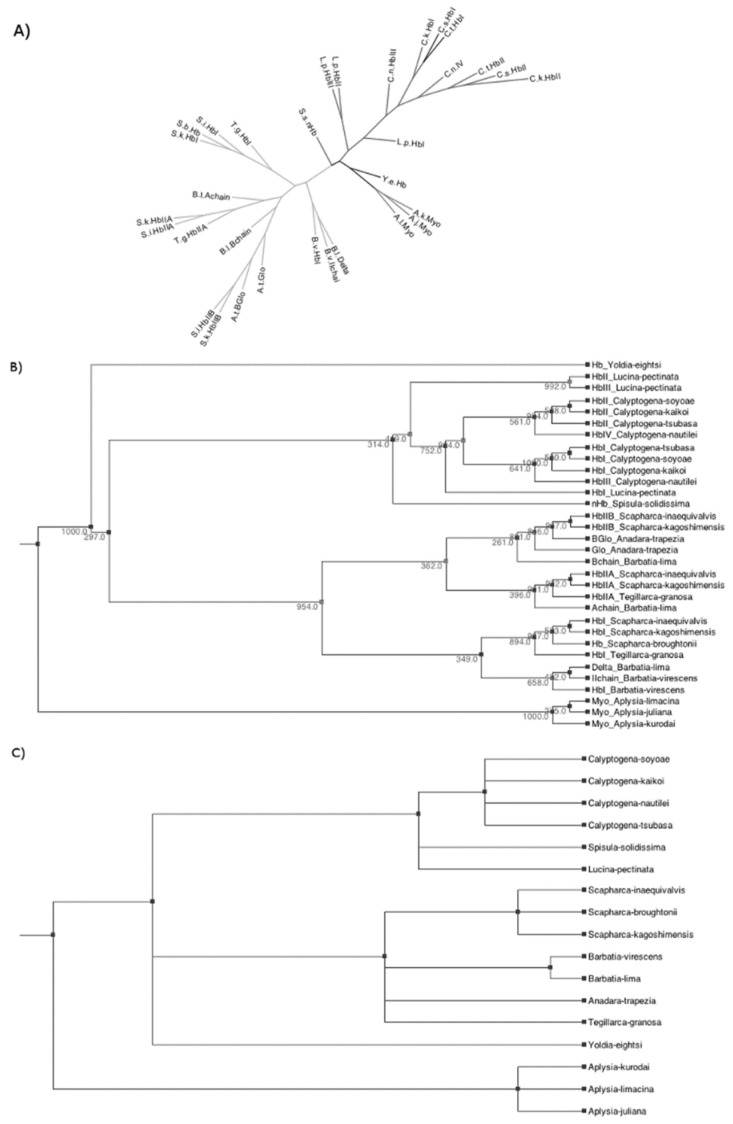
(**A**) Maximum-likelihood tree of based on 32 mollusk globin sequences with 1000 bootstraps (bootstrap values are shown in the reconciled tree). (**B**) Gene Tree. A rooted-species-reconciled phylogenetic gene tree was constructed using PHYLIP, followed by reconciling the gene tree with the species tree and rooting using NOTUNG, as described in Materials and Methods. Numbers represent the bootstrap values from the Maximum Likelihood phylogenetic analysis. (**C**) Species tree. A rooted species phylogenetic tree was built using the Taxonomy resource at NCBI and verified using the Tree-of-Life resource at http://tolweb.org (accessed on 12 November 2018).

**Figure 4 genes-13-02041-f004:**
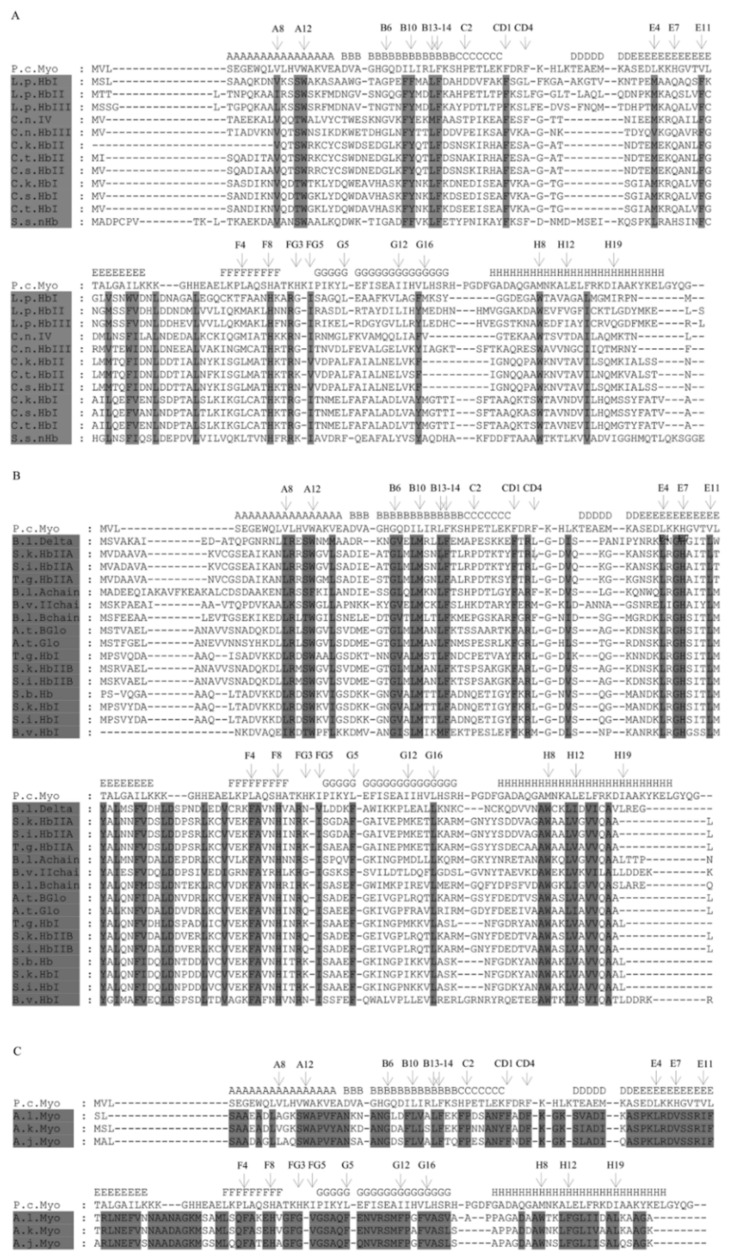
Visualization of Multiple Sequence Alignment showing amino acid residues that are conserved between each clade. (**A**) Bivalves belonging to the Heterodonta subclass. (**B**) Bivalves belonging to Pteriomophia subclass. (**C**) Gastropods belonging to Opisthobranchia subclass. For abbreviations see legend of Figure 2. Above each of the alignments, we indicate the secondary structure of *P. catodon* Myoglobin according to its crystallographic structure (accession number 1EBC_A). The alignments were visualized using GeneDoc.

**Figure 5 genes-13-02041-f005:**
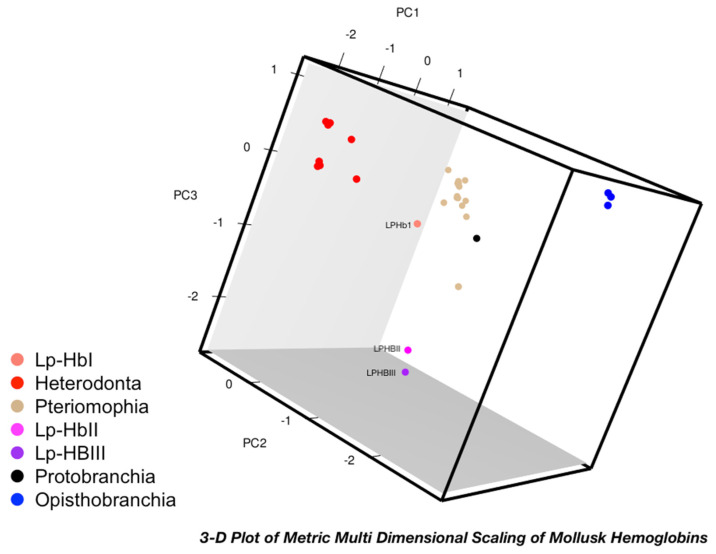
3D representation of the mollusk globins sequence space. A trimmed multiple sequence alignment of 32 homologous mollusk globins was analyzed by MMDS, with distances based on the JTT distance matrix. The 3D space is defined by the first three components of the MDS analysis. HbI_Lp_ is colored ‘salmon’, HbII_Lp_ is colored in magenta and HbIII_Lp_ is colored purple. The rest of the sequences are color coded according to mollusk sub-families they belong to: colored in red are Heterodonta subclass, in tan are the Pteriomophia subclass, in blue are the Opisthobranchia subclass, and in black the Bivalve, *Y. eightsi*, member of the Protobranchia subclass.

**Figure 6 genes-13-02041-f006:**
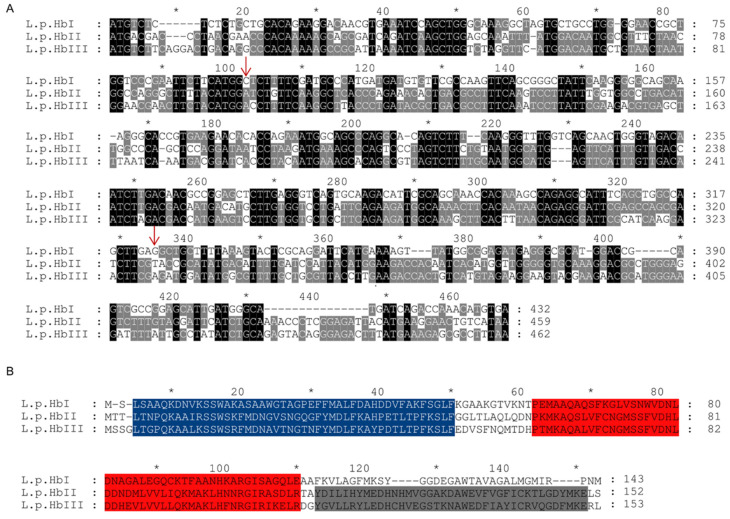
(**A**) Multiple sequence alignment and conservation levels of the coding regions of hemoglobins from *L. pectinata***.** Conservation levels between the three coding region sequences of HbI_Lp_, HbII_Lp_ and HbIII_Lp_ of *L. pectinata* were visualized using GeneDoc. Nucleotides shaded in black represent 100% of conservation, in gray, 66% conservation, and nucleotides having less than 66% conservation were not shaded. Red arrows indicate position of intron insertions. (**B**) Multiple sequence alignment of *L. pectinata* hemoglobins. MSAs were carried out using M-COFFEE and visualized in GeneDoc. Motif analysis was performed using MEME and identified motifs are represented in different colors.

**Figure 7 genes-13-02041-f007:**
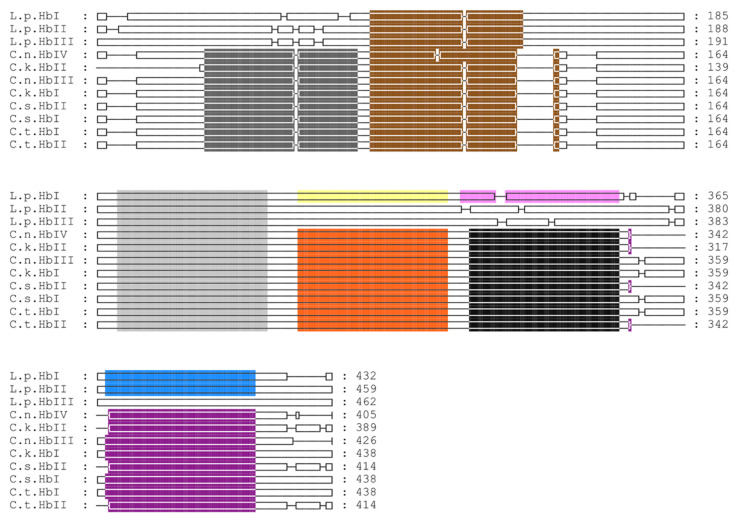
Multiple sequence alignment of coding regions of hemoglobins from Lucina pectinata and Calyptogena. Multiple sequence alignments were carried out using M-COFFEE, motif analysis was done using MEME and aligned sequences visualized in GeneDoc. The abbreviations are as follows: L.p. (*L. pectinata*), C.n. (*C. nautilei*), C.k. (*C. kaikoi*), C.t. (*C. tsubasa*), C.s. (*C. soyoae*). Detected motifs are represented in different colors.

**Table 1 genes-13-02041-t001:** Amino acids in positions conserved in organism living in sulfide-rich environments.

Organism	Globin	Abbreviation	Amino Acids in Positions:			
			B9	B10	E7	E11
*L. pectinata*	Hemoglobin I	HbI_Lp_	Phe	Phe	Gln	Phe
*L. pectinata*	Hemoglobin II	HbII_Lp_	Phe	Tyr	Gln	Phe
*L. pectinata*	Hemoglobin III	HbIII_Lp_	Phe	Tyr	Gln	Phe
*C. nautilei*	Hemoglobin IV	HbIV_Cn_	Phe	Tyr	Gln	Phe
*C. nautilei*	Hemoglobin III	HbIII_Cn_	Phe	Tyr	Gln	Phe
*C. kaikoi*	Hemoglobin II	HbII_Ck_	Phe	Tyr	Gln	Phe
*C. tsubasa*	Hemoglobin II	HbII_Ct_	Phe	Tyr	Gln	Phe
*C. soyoae*	Hemoglobin II	HbII_Cs_	Phe	Tyr	Gln	Phe
*C. kaikoi*	Hemoglobin I	HbI_Ck_	Phe	Tyr	Gln	Phe
*C. soyoae*	Hemoglobin I	HbI_Cs_	Phe	Tyr	Gln	Phe
*C. tsubasa*	Hemoglobin I	HbI_Ct_	Phe	Tyr	Gln	Phe

## Data Availability

Not applicable.

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
