# Peer review of "Bioinformatic Characterization and Molecular Evolution of the Lucina pectinata Hemoglobins"

_genes, 2022, doi:10.3390/genes13112041_

Round 1

Reviewer 1 Report

The authors have put together a clearly written article describing the structure and molecular evolution of L. pectinata hemoglobins. The analyses all seem relevant and are well explained.  I think need some minor revisions, detailed below:

Abstract

Line 20: misspelled “hemogloblin”

Introduction

Lines 36-37: In invertebrates?

Figure 1B: I’m not convinced that you need to focus on the similarity with sperm whale myoglobin here. I think this figure would be sufficient with just the 1A part.

Materials & Methods

Line 102: Describe how you acquired the sequences for L. pectinata hemoglobins.

Line 117: Please explain why you are comparing your bivalve sequences to sperm whale myoglobin.

Line 144: Why is this information here and not in the beginning of your Methods section?

Results

Genus and species names should be italicized

Lines 165-174: this reads as if it should be in your Methods section, not in the Results. Please describe the results of your alignment here instead.

Figure 2 & 3: Please choose colors that do not include red and green since these can be hard for some color blind individuals to see. Please apply this comment to all subsequent figures as well.

Line 214: Major clades?

Figure 5: I suggest keeping only the 3D MMDS in the manuscript and moving the others to the supplement as it is redundant to include all these plots. I think it would make interpretation of the 3D plot smoother if you labeled the Lp hemoglobins in the plot.

Discussion

Line 347: change to “than the one”

Line 372: Please explicitly explain the significance of this statement.

Generally, the Discussion lacks contextualization of statements. A lot of links are made across studies and species, but there are no statements or hypotheses suggesting why these links exist. The Discussion would be a more compelling, and seemingly relevant, read if this type of information is added. Most of this information is in the Conclusion paragraph, but I would prefer to see it distributed throughout the Discussion, followed by a more succinct Conclusion.

Reviewer 2 Report

Reviewer comments 

The manuscript probes into the evolutionary characteristics of three Hemoglobins (HbI, HbII and HbIII) found in the mollusc Lucina pectinata. L. pectinata is found in sulfide-rich environments. Hemoglobin is often considered to be a carrier of oxygen. However, of the three types of hemoglobins found in L. pectinata, HbI is considered responsible for the absorption and transport of hydrogen sulfide, whereas HbII and HbIII transport oxygen. The question then arises how and what events during evolution led to the origin of HbI. Through detailed bioinformatics and phylogenetic analysis, the authors investigate the events that led to the origin of HbI, HbII and HbIII in L. pectinata. Through their study, the authors have reported two very interesting findings. First, HbI is originally a close homolog of the hemoglobins found in species of Calyptogena, which are often found near hydrothermal vents of oceans. Second, HbI originated through a gene duplication event in the common ancestor of HbI, HbII and HbIII. The event that led to HbII and HbIII existing as separate molecules was a more recent duplication event on the evolutionary scale. The manuscript is very well written, and the analysis (based on the reviewer’s understanding) is thorough. The authors have also complemented their findings through principal component analysis that shows HbI clustered separately from HbII and HbIII. I only have a few minor comments/questions.

 1.  Often the multiple sequence alignment can be a big factor in determining the topology of the phylogenetic tree. I noticed that the authors have used PSI-COFFEE for performing the multiple sequence alignment. Why not use other popular methods, such as Clustal Omega or MAFFT? Do any of them (MAFFT/Clustal Omega) give the same alignment that leads to the same gene tree (Fig. 3B)?

2.     Figure 2 could be improved in terms of its resolution and representation. The transparency of the cartoons could be reduced to illustrate the extent of structural similarity between the superimposed proteins. What was the RMSD for the aligned structures?
